# Superhydrophobic Coating Derived from Geothermal Silica to Enhance Material Durability of Bamboo Using Hexadimethylsilazane (HMDS) and Trimethylchlorosilane (TMCS)

**DOI:** 10.3390/ma14030530

**Published:** 2021-01-22

**Authors:** Silviana Silviana, Adi Darmawan, Febio Dalanta, Agus Subagio, Ferry Hermawan, Hansel Milen Santoso

**Affiliations:** 1Department of Chemical Engineering, Diponegoro University, Jl. Prof. H. Soedarto, SH., Semarang 50275, Indonesia; hanselmilensantoso@students.undip.ac.id; 2Department of Chemistry, Diponegoro University, Jl. Prof. H. Soedarto, SH., Semarang 50275, Indonesia; adidarmawan@live.undip.ac.id; 3Master Program of Chemical Engineering, Diponegoro University, Jl. Prof. H. Soedarto, SH., Semarang 50275, Indonesia; dalanta@student.undip.ac.id; 4Department of Physics, Diponegoro University, Jl. Prof. H. Soedarto, SH., Semarang 50275, Indonesia; agussubagio@lecturer.undip.ac.id; 5Department of Civil Engineering, Diponegoro Univesity, Jl. Prof. H. Soedarto, SH., Semarang 50275, Indonesia; ferry.hermawan@live.undip.ac.id

**Keywords:** bamboo coating, geothermal silica, statistical, HMDS, TMCS

## Abstract

Bamboo, a fast-growing plant from Asia, is used as building material with unique properties, while exhibiting fast degradation due to its hydrophobicity. Therefore, many attempts have been implemented using several technologies for bamboo modification to alter the hydrophobicity. Most previous studies producing superhydrophobic properties are conducted by using tetraethoxysilane (TEOS) as a precursor agent. However, this method, using TEOS with harmful properties and unaffordable compounds, requires many steps to accomplish the experimental method. Therefore, this paper employed geothermal solid waste as a silica source of the precursor. Thus, an effective and efficient method was applied to prepare superhydrophobic coating by using a precursor of geothermal silica and further modification using hexamethyldisilazane (HMDS) and trimethylchlorosilane (TMCS). The research was executed by the full factorial statistical method using two numerical variables (HMDS/TMCS concentration and silica concentration) and one categorical variable (solvent types). The uncoated material revealed higher weight gain in mass and moisture content than that of the coated bamboo after the soil burial test to assess the durability of the bamboo. However, the durability of superhydrophobic coating realized hydrophobic performance for both agents during sand abrasion for a total of 120 s at an angle of 45°. Statistical results showed the optimum contact angle (CA) achieved in superhydrophobic performance with lower silica concentration for HMDS concentration and the appropriate solvent of n-hexane for HMDS and iso-octane for TMCS. All results were supported using many instruments of analysis to confirm the step-by-step alteration of geothermal silica to be used as a superhydrophobic coating, such as XRF, XRD, FTIR, SEM, and SEM EDX.

## 1. Introduction

High humidity in most tropical countries causes the degrading of biomaterials by means of the decline in the nature and function of materials. The humidity can change material characteristics both in terms of physical and chemical properties [1]. The material characteristics in conditions of high humidity can be characterized by corrosion in metal materials, condensation on the surface of glass materials promoting growth of fungi and bacteria, and very fast degradation of bamboo materials [2]. Therefore, many methods have been implemented to maintain the properties and functions of those materials. The common method refers to an application of a coating.

In metal materials, coatings are carried out by application on any metals with a relatively high thickness of 8 mm [3]. In addition, coating on metal is supported by alloy materials such as stainless steel with high corrosion resistance, and the associated costs are affordable [3]. For other material, such as bamboo material, the coating method is to put on paint/varnish or by dipping/soaking in boric acid for several days. However, the coating on bamboo material is normally easily peeled off [4]. Consequently, many of these coating methods are not satisfactory.

Material coating methods have been developed to overcome these weaknesses. This method is considered to be effective through the use of a superhydrophobic coating method to maintain material quality. This method focuses on preparation of a superhydrophobic coating to protect material from material degradation. Moreover, the self-cleaning properties produced can resist water and carry dirt on the surface of the material. Many studies have been executed on superhydrophobic coatings such as with the use of fluorocarbon compounds which only produce a contact angle (CA) of 110°. As the superhydrophobicity is achieved by the combination of low surface energy materials and micro/nanoscale texture, most artificial superhydrophobic materials cannot withstand the mechanical abrasion due to the destruction of micro/nano structures. Multi-fluorination strategy based on fluorinated epoxy resin can improve the mechanical and chemical properties of superhydrophobic materials [5]. Perfluoropolyether and fluoro polymeric nanoparticles are used as their building blocks. The epoxy resin is selected due to its mechanical and chemical robustness, ability to disperse nanoparticles through hydrophilic functional groups and strong substrate adhesion; the perfluoropolyether helps tune the surface energy and flexibility; and the fluoropolymer nanoparticles offer the texture control and low surface energy [6]. Research in the manufacture of hydrophobic coatings using fluoroalkylsilane-type compounds such as fluoroalkylsilane (FAS) [7] and heptadekaflorodesyltrimethoxysilane (HFTS) has been carried out [8]. However, the use of these compounds has a negative impact on health and the environment. Furthermore, hydrophobic coating research has converted to the use of non-fluoro compounds that are less toxic like hexadecyltrimethoxysilane (HDTMS), hexamethyldisilazane (HMDS) and octadecyltriklorosilane (ODTCS) and trimethylchlorosilane (TMCS) [9].

HMDS (hexamethyldisilazane) is a colorless liquid and popularly used as a reagent and a precursor to bases in organic synthesis and organometallic chemistry. HMDS was obtained from ammonia derivatives with trimethylsilyl groups in place of two hydrogen atoms. In addition, HMDS can be synthesized from TMCS [10]. HMDS treatments could exert a good influence on increasing the crystallinity resulting in the increase of materials resistance with water [11]. Therefore, HMDS can be used as a coating material. 

The similar research finding substitution of silica sources can be further observed. The potential source of raw material refers to geothermal silica to replace water glass and TEOS as a source of silica in the superhydrophobic coating in this study. The addition of water glass requires two stages of immersion and the use of tetraethoxysilane (TEOS) can produce a contact angle of 150−180° but it causes inhalation toxicity [12]. Previous research results show that geothermal scaling waste has a silica content of up to 88% and a high silica content will increase surface hydrophobicity while having a lower level of toxicity than TEOS [12,13]. Therefore, silica obtained from geothermal scaling waste is very potential to be used as raw material for superhydrophobic coating to replace TEOS. Geothermal solid waste has the potential as a sustainable source of silica due to high silica content and the capacity for a huge amount of waste. The Indonesian Institute of Sciences (LIPI) stated that geothermal waste in Dieng has a liquid waste as brine water capacity of 10 tons/day releasing solid waste of one ton/day [13]. This paper investigates the preparation of a superhydrophobic layer using geothermal solid waste as a precursor of sodium silicate with the addition hexamethyldisilazane (HMDS) and trimethylchlorosilane (TMCS) through a single stage mixing process by means of a spray coating method. The investigation used statistical studies using factorial design with three variables of two numerical variables and one categorical variable, i.e., HDMS/TMCS concentration, silica concentration, and solvents type with response of contact angle. This study compared the utilization of each surface-modifying agent, TMCS and HMDS, on bamboo surfaces which are susceptible to damage due to air humidity. Furthermore, durability of the bamboo was also observed by a soil burial test to examine the weight gain and moisture content of the samples. Meanwhile, testing of the durability of the superhydrophobic coating was conducted using the sand abrasion test [14].

## 2. Materials and Methods

### 2.1. Preparation and Purification of Geothermal Silica

Geothermal silica waste obtained was treated by drying at 110 °C for 4 h to remove water content. Then, the dried geothermal silica was reduced to a particle size of 400 nm using high-energy milling (HEM). Inorganic impurities were removed by mixing 200 mL of hydrochloric acid with 6 mole/L by heating for 3 h at 90 °C. Then, washing the solution obtained neutral pH. The filtration was conducted to separate the solid silica and filtrate. The solid was subjected to drying for 3 h. The dried silica was introduced by sieving and releasing nano-sized silica by high-energy milling for 2 h [13].

### 2.2. Superhydrophobic Solution Synthesis

Superhydrophobic solution was made by dissolving nanometer-sized silica at various concentrations in 20 mL of each solvent, i.e., n-hexane (≥95%, Merck, Darmstadt, Germany), ico octane (≥99%, Singapore-Sigma Aldrich), cyclohexane (≥99%, Singapore-Sigma Aldrich), and xylene (≥99%, Singapore-Sigma Aldrich). Then, TMCS (≥99%, Merck, Darmstadt, Germany) or HMDS (≥97%, Wacker Chemi AG, Burghausen, Germany)was added at various concentrations slowly into silica solution at 50 °C for 2 h [15]. Subsequently, the solution was applied to the material using a commercial plastic sprayer by 0.05 mL in volume for each spraying. The commercial sprayer has length of pipe of 8 cm, length of spring of 1.2 cm, diameter of spray bottle of 2 cm, and height of spray bottle of 6 cm. The spray was used 3 times on each surface of the samples to ensure the coating was spread homogeneously. The distance between sprayer and the sample was maintained at 25 cm. This research was statistically conducted by using full factorial design of 3 variables with 3 levels of each variables (Table 1). Therefore, each experiment was accomplished with 27 runs. The ANOVA result was obtained based on Design Expert 8.0.6 software.

### 2.3. Product Characterization

Characterizations of coated material was carried out by use of Race Contact Angle to obtain contact angle, analysis of Scanning Electron Microscopy (SEM FEI Type Inspect-S50), X-ray Fluorescence (XRF Panalytical Type Minipal 4, Malvern, PA, USA), X-ray Diffraction (XRD) Panalytical type Expert Pro, Malvern, PA, USA, Fourier Transform Infra-Red (FTIR) IRPrestige21, Shimadzu, Chiyoda-ku, Tokyo, Japan, and Scanning Electron Microscopy Energy Diffraction X-rays (SEM EDX Thermo Fischer Scientific, Waltham, MA, USA).

### 2.4. Bamboo Durability Test

In the material durability test, samples of 4 cm × 4 cm of coated bamboo and uncoated bamboo samples were buried in soil in a desiccator with adjusted relative humidity by using supersaturated KCL solution to maintain 85% relative humidity [16] for 10 weeks. Every day the soil was watered to keep humidity. Every week, the buried samples were cleaned up from the soil and weighted. The moisture contents and weights of the samples before and after the soil burial test were recorded for 10 weeks. This procedure adopted the procedure of previous research with slight modification according to the purpose of this study [17].

### 2.5. Durability of Superhydrophobic Coating Test

The test was conducted to observe the durability of the superhydrophobic coating [14]. The test applied sand with particle size of 35 µm. The coated surface of bamboo sample was mounted at angle of 45° below a sieve. The sieve was used to spread affected area with sand by the dropping of sand above 10 cm from the sample. Each sample was introduced to 30 to 120 s of sand abrasion at an angle of 45°, then the contact angles of the samples were observed. This procedure referred to other research by mounting a 45° angle [14].

## 3. Results and Discussions

### 3.1. Morphology, Size, and Surface Area of Silica

Based on previous research [13], the surface morphology of geothermal silica waste has an increase in pore uniformity of 30% to a porous structure with a pore uniformity of 90% [13]. Pore uniformity affects silica dissolution wherein pore uniformity above 80% can prevent the agglomeration of superhydrophobic silica solution [10]. The silica in this solution is nanometer-sized and is able to be dissolved evenly based on the results of the PSA (Particle Size Analysis) analysis (Panalytical, Malvern, PA, USA) analysis. In Table 2, the surface area prior to treatment was 40.90 m^2^/g, and it rose sharply to 125.56 m^2^/g after treatment [13]. Large surface area has been found to increase silica reactivity [18].

### 3.2. Mineral Contents in Geothermal Silica Scaling Waste

The contents of Fe and Mn components in silica can affect the physical properties such as the surface area, porosity, and particle size. Removal of impurities in silica was executed through an acid leaching method with hydrochloric acid. The hydrochloric acid can dissolve impure metal elements and increase the surface area of silica [10]. Table 2 shows that silica content increased from 86.3% to 97.1%. This evidence revealed the effectiveness of the purification process as the nanometer-sized silica was created making the transfer of mass through hydrochloric acid and chemical bond breaking in silica easier [19].

XRF analysis from geothermal silica before and after acid the leaching process are shown in Figure 1 and Figure 2. 

### 3.3. Structure of Geothermal Silica Scaling Waste Material

Determination of the crystalline structure of silica scaling used in the manufacture of superhydrophobic solutions was executed by XRD analysis. Silica (SiO_2_) after pretreatment had the steepest peak of 22.64° and an intensity count of 185 as shown in Figure 3. Based on the data, silica scaling before pretreatment has a crystallinity of 40% [9] whereas after treatment it has a crystallinity of 20% (Figure 3). This is due to the purification and milling processes carried out that are capable of breaking the silica crystal structure into irregular (amorphous) structures. Amorphous silica has a small and porous crystalline size providing mass transfer during the reaction process with HMDS in non-polar solvents [20].

### 3.4. Analysis of Hydrophobicity, Contact Angle, and the Determination of Optimum Conditions

To determine the optimum conditions from this experiment, researchers used Design Expert 8.0.6 software for data processing, statistical approaches, and the optimization of the variables. The experiment result consisted of 27 runs which was processed with the full factorial method consisting of two numerical variables (HMDS/TMCS concentration and silica concentration) and one categorical variable (solvent types). The contact angle responses of all the runs are shown in Table 3 and Table 4.

#### 3.4.1. Effect of HMDS Concentration on Contact Angle

The effect of the surface-modifying agent on the wettability properties was analyzed by varying the concentration of HMDS (1.5%-*v*/*v*; 4%-*v*/*v*; 6.5%-*v*/*v*) with n-hexane. When the silica concentration used 3%-*w*/*v*, increase in HMDS concentration resulted in greater contact angles as shown in Figure 4. The best HMDS concentration obtained is 6.5%-*v*/*v* with maximum contact angle achieved with n-hexane solvent. This presents the successful exchange of silanol groups with alkyl on the surface of silica. At low concentrations, the resulting contact angles do not meet the hydrophobic criteria. This is because a low concentration of HMDS causes the reaction to occur in the diffusion-limited region with the reaction mechanism in Equations (1) and (2) [21].
2(CH_3_)_3_SiCl + H_2_O → (CH_3_)_3_Si–O–Si(CH_3_)_3_ + 2HCl(1)
(CH_3_)_3_SiCl + Si–OH → (CH_3_)_3_Si–O–Si + HCl(2)

When n-hexane was used as solvent, the HMDS concentration of 6.5%-*v*/*v* produced a maximum CA with a silica concentration of 3%-*w*/*v*, whereas when isooctane was used as solvent, the HMDS concentration of 1.5%-*v*/*v* produced a maximum CA with a silica concentration of 3%-*w*/*v*; when cyclohexane was used with a 5.5% silica concentration, the maximum CA would be produced with 1.5% HMDS. These results can be seen in Figure 4, Figure 5 and Figure 6, respectively. Figure 4, Figure 5 and Figure 6 visualize the interaction graph when a different response is obtained with the two adjusted factors (variables).

Optimum CA was statistically obtained as 161.9° with the use of HMDS at a concentration of 6.5%, silica concentration of 3% and n-hexane solvent upon a desirability of 100%, as shown in Figure 7. Figure 7 reflects a sequence of graphical views (in ramps) of each optimal solution. Red points denote optimal factors/variables, while a grey point shows responses with the chosen statistical model. Moreover, a blue point depicts the optimal response prediction. In optimization criteria, each factor/variable can choose goal in range of the experiment conditions, while response (contact angle) can be adjusted goal to be maximize of the experimental conditions

#### 3.4.2. Effect of Type of Solvents used on Contact Angle

The appropriate solvent was determined by varying the types of solvents based on the differences in their chemical structure. The solvents used n-hexane, isooctane and cyclohexane with silica and HMDS concentration variations (Figure 5, Figure 6 and Figure 7). However, the maximum CA was achieved by using n-hexane as solvent, HMDS concentration of 6.5%, and silica concentration of 3% as HMDS has less polarity as a silylation agent. Moreover, HMDS can be generated from TMCS [8]. Therefore, silyl derivatives are generally more volatile, less polar, and more thermally stable. Table 5 shows that n-hexane has a lesser polarity than that of other solvents [22].

The solvent exchange occurs between the alkyl group in solvent to the silane agent TMCS. The solvent type in solvent exchange can be affected by functional group of hydroxyl (O–H). It was supported from previous research that using isooctane solvent with TMCS 13%-*v*/*v* produces a contact angle of 179°, confirmed by low peak of O–H spectra [23].

One of the samples with HMDS was analyzed by FTIR. The chemical functional groups attached on the silica after surface modifications were evaluated using FTIR (Figure 8). The unmodified silica sample showed a broad peak at around 3300 cm^−1^; the existence of a huge fraction of elemental Si was confirmed as the hydroxylated species. This also corresponds to the O–H stretching peak as water molecules; the peak at 1630 cm^−1^ denotes –OH groups and at 960 cm^−1^ it corresponds to Si–OH. Both of these results were confirmed for the unmodified silica [22], while the modified silica using TMCS showed the peaks at 2960 cm^−1^ and 850 cm^−1^ attributed to Si–CH_3_ bonding and stretching, respectively [22]. These peaks were confirmed as the effect of TMCS modification on surface of the silica. The surface modification reaction between TMCS and silica with n-hexane provides the replacement of Si–OH groups (hydrophilic) to Si–CH_3_ groups (hydrophobic) which can be confirmed by the appearance of new peaks at 2960 and 850 cm^−1^ [23,24]. From the FTIR spectrum of modified silica using HMDS, it can be interpreted that the sharp peaks found at 2966 cm^−1^ and 2932 cm^−1^ indicate the C–H stretching bonding in –OSi(CH_3_)_3_ resulting from the reaction with HMDS. Moreover, the peak at around 1400 cm^−1^ was assigned to the N–H bonding as well as possibly to the vibration of CH_2_ group due to the modification using HMDS [25]. The asymmetric vibration peak at 1100 cm^−1^ in all samples indicated the vibrations of Si–O–Si bonding [25,26].

Based on the FTIR result, it was confirmed that the use of TMCS to modify a silica coating produced a higher wave transmittance than that of HMDS. This analysis result indicated that silica coatings with TMCS generated more superhydrophobic properties than that of HMDS addition. TMCS is well known as a stronger silylation agent than HMDS [27]. The addition of TMCS in isooctane solvent released a higher contact angle response from the material of bamboo.

The superhydrophobic coating derived from silica using TMCS or HMDS as surface modifying agenprovided the more significant transmittance, which was caused by the larger formation of a silica network (Si–O–Si) after the modification with silane agent. Based on these FTIR spectra, therefore, this modification has been confirmed to facilitate the significant transformation of hydrophobic behavior on the silica surface.

#### 3.4.3. Effect of Silica Concentration on the Contact Angle Produced

Figure 9 and Figure 10 visualize the effect of silica concentrations on the contact angle with various types of solvents. At the optimum silica concentration of 3%-*w*/*v*, the highest contact angle is achieved 161.9°, an angle which is within the hydrophobic and self-cleaning criteria of bamboo material containing 3%-*w*/*v* silica concentration in n-hexane solvent as shown in Figure 9. The test was carried out with a fixed variable of 6.5%-*v*/*v* HMDS. Meanwhile, Figure 10 gives evidence that at 13%-*v*/*v* TMCS generates an increase in the concentration of silica resulting in the highest contact angles with the isooctane solvent.

Based on Figure 9 and Figure 10, it can be seen that using TMCS required more than using HMDS as surface modifying agent to achieved in superhydrophobic properties. The amount of TMCS is required theoretically twice as much as that of HMDS as surface modifying agent (Figure 11). 

Figure 12 shows that the use of TMCS 3% with n-hexane solvent released a maximum CA of 144° with 5.5% silica. Whereas in Figure 13, a contact angle of 162° was obtained with isooctane solvent using TMCS 13% with a silica concentration of 5.5%. The use of xylene solvent produced a contact angle of 151° (Figure 14).

Optimum CA was achieved at 166.9° with the use of TMCS at a concentration of 13%, silica concentration of 5.5% and iso-octane solvent that reached a desirability of 87% as shown in Figure 15.

#### 3.4.4. Durability Test on Bamboo

Wet bamboo can be appropriate for the growth of various fungi. Meanwhile, coated bamboo can prevent moisture content from infiltrating into the bamboo pers, despite requiring only a slight mass increase. In this paper, soil burial testing of coated bamboo with hydrophobic and superhydrophobic bamboo in the soil was assessed.

Table 6 shows that coated bamboo with has a mass increase of 9.5% and 26% moisture content while uncoated bamboo attained 155%-w increase in mass and 28% MC (moisture content) after soil burial testing for 10 weeks. Moreover, it was also assessed that coated bamboo had 112.5° (hydrophobic) and 180° (superhydrophobic) contact angles after 10 weeks with modified silica with HMDS.

It can be seen in Table 7, durability test resulted 12.9%-w of mass increase and 21% of moisture content increase for hydrophobic coated sample, while the superhydrophobic coated sample released 2.6% of mass increase and 19% of moisture content increase. The coated bamboos with HMDS introduced in the assessment had a thickness of 7.49 mm, while the coating thickness was around 55–66 µm for a 142° contact angle and around 94.7 µm a 180° contact angle; this can be seen in Figure 16. Figure 16 reflects two surfaces of two different samples of HMDS (6.5% and 4%), silica (3%), and solvents of n-hexane and isooctane, respectively. The sample of 4% HMDS and 3% silica showed a contact angle of 112.5°, while 6.5% HMDS and 3% silica gave a contact angle of 180°. Due to the significant difference in the contact angles, the surface analysis suggests that a higher contact angle can be obtained with more roughness than with a lower contact angle. The hydrophobicity can be inferred by the surface roughness and silane agent ratio [29]. Moreover, the smoother surface, such as in Figure 16b, implies that different solvent exchange can affect the roughness due to the level of the crystallinity depending on how much HMDS is dissolved in the superhydrophobic solution [9].

SEM images in Figure 17 and Figure 18 can distinguish before and after modified bamboo with the coatings; moreover, surface analysis has been examined by using SEM EDX mapping to obtain surface morphology including the elemental content on the surface. Figure 19 reflects two surfaces of two different samples TMCS (8% and 3%), silica (0.5% and 5.5%), and solvents of n-hexane and iso-octane, respectively. For 8% TMCS and 0.5% silica, a contact angle of 129° was produced, while 3% TMCS and 5.5% silica exhibited a contact angle of 79°. The hydrophobicity can be obtained by the surface roughness and silane agent ratio [30].

#### 3.4.5. Durability Test of Superhydrophobic Coating

The contact angle during the sand abrasion test was recorded to assess the durability of the superhydrophobic coating. It can be seen in Table 8 below.

Both samples inferred that after 2 min of abrasive processing, the sample shows significant abrasion delustering and complete wetting by water [31]. The abrasion process is accompanied with local deformations of the surface and thus with the increase in surface energy. This phenomenon is known as the thermo-mechanical activation of the surface. That is why, just after abrasion treatment, the sample surface state is characterized by high non-equilibrium with enhanced adsorption activity. Such adsorption activity on the surface of thermo-mechanically activated materials is driven by the tendency to reduce the total free (Gibbs) energy of the system by diminishing the surface energy part. Moreover, the contact angle of samples after the sand abrasion test can be seen in Figure 20.

## 4. Conclusions

A self-cleaning superhydrophobic nano-coating has been successfully made using an amorphous silica structure in nanoparticle silica with a specific surface area of 125.56 m^2^/g with 97.1% silica content. The increase in silica concentration and HMDS can increase the contact angle using n-hexane solvent. Statistical results on bamboo material achieved a contact angle of 161.9° with 100% desirability by use of silica concentration of 3%-*w*/*v*, 6.5%-*v*/*v* HMDS and n-hexane solvent onto bamboo material. Meanwhile, a contact angle of 166.9° was achieved with the use of TMCS at a concentration of 13%, silica concentration of 5.5% and isooctane solvent. However, coated bamboo with HMDS revealed lower mass gain and moisture content after burial test in soil for 10 weeks. After the sand abrasion test, the coated samples can attain hydrophobic properties due to the decrease in the contact angle below 150°.

## Figures and Tables

**Figure 1 materials-14-00530-f001:**
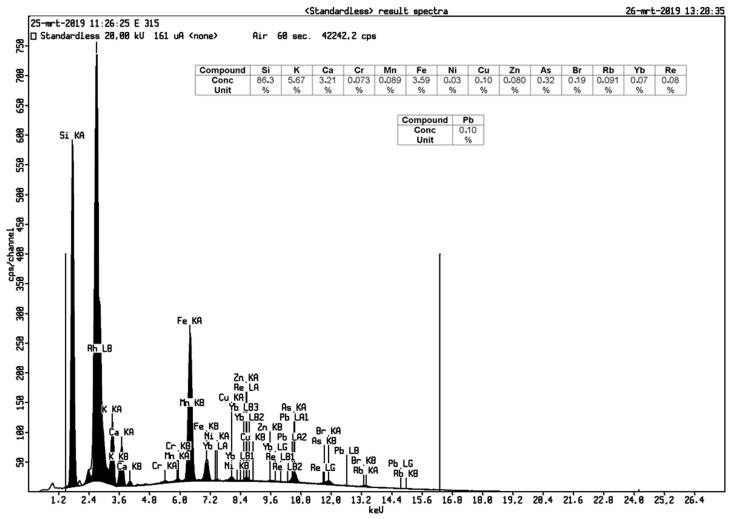
XRF analysis of geothermal silica before acid leaching.

**Figure 2 materials-14-00530-f002:**
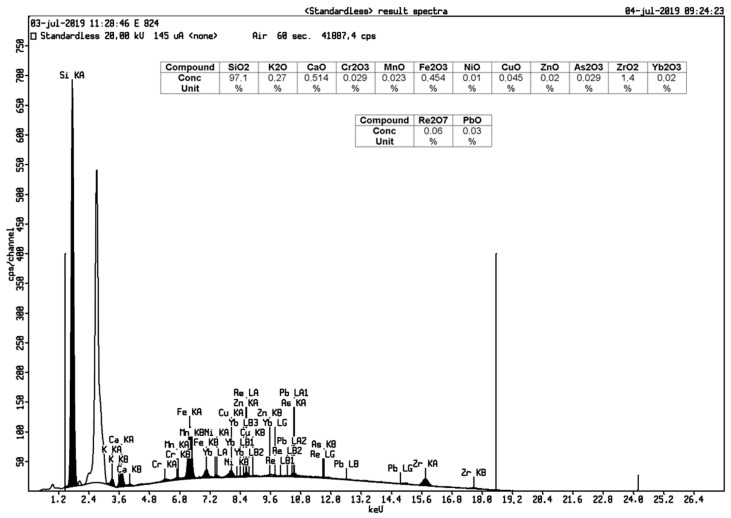
XRF analysis of geothermal silica after acid leaching.

**Figure 3 materials-14-00530-f003:**
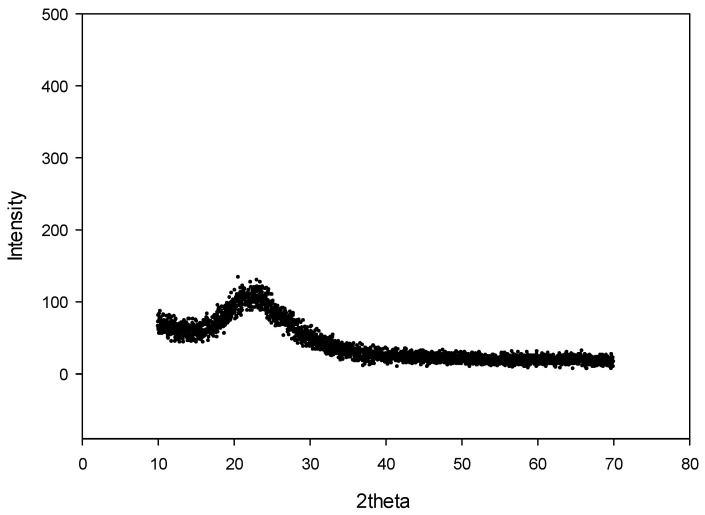
XRD analysis of geothermal silica after acid leaching.

**Figure 4 materials-14-00530-f004:**
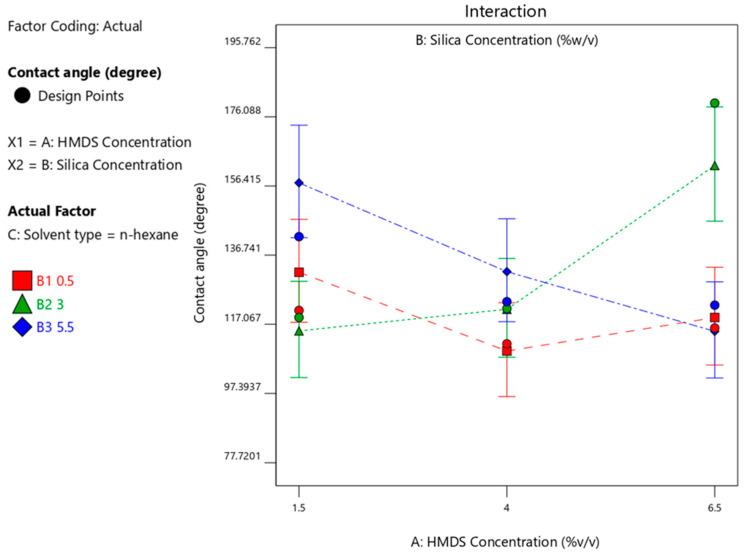
Contact angle response upon interaction between HMDS and silica concentration with n-hexane solvent.

**Figure 5 materials-14-00530-f005:**
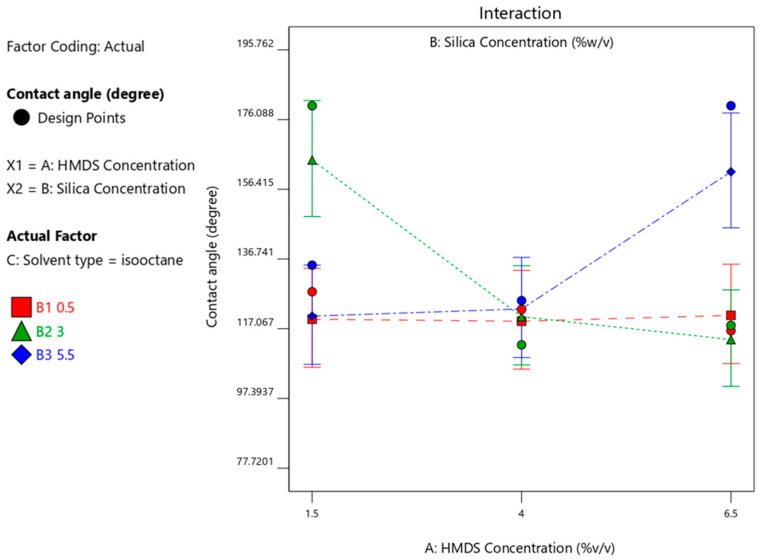
Contact angle response upon interaction between HMDS and silica concentration with isooctane solvent.

**Figure 6 materials-14-00530-f006:**
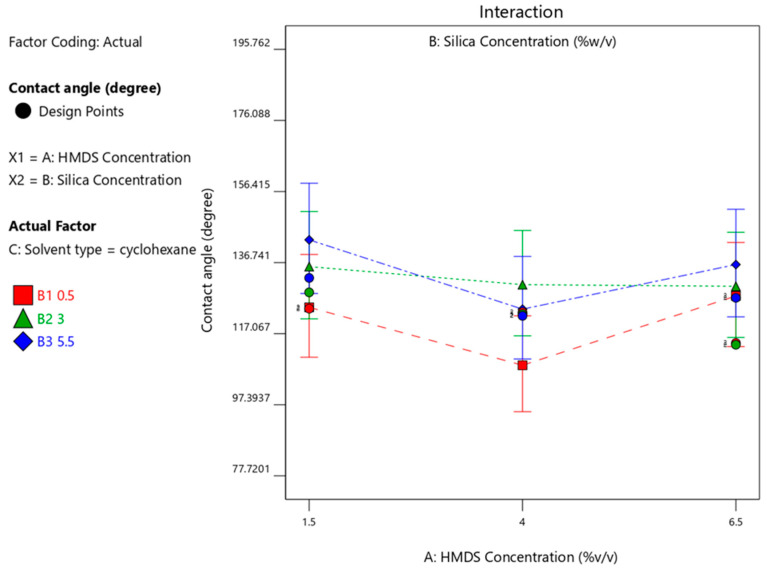
Contact angle response interaction between HMDS and silica concentration with cyclohexane solvent.

**Figure 7 materials-14-00530-f007:**
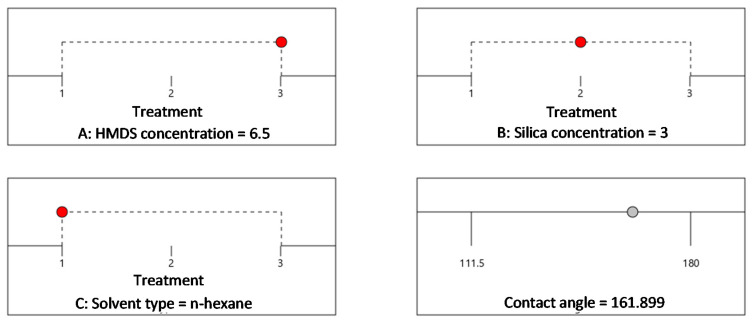
Optimization of the operating conditions of superhydrophobic coating using HMDS.

**Figure 8 materials-14-00530-f008:**
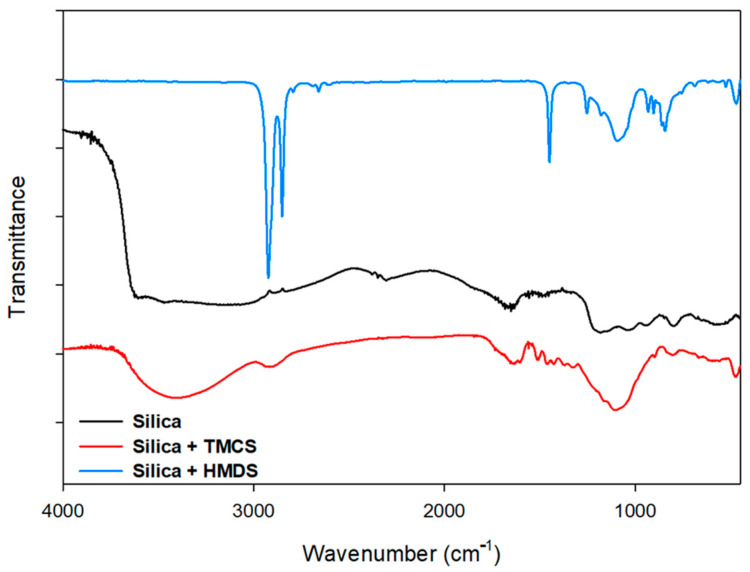
FTIR spectra of silica, superhydrophobic silica dissolved in solvent of n-hexane with TMCS and HMDS.

**Figure 9 materials-14-00530-f009:**
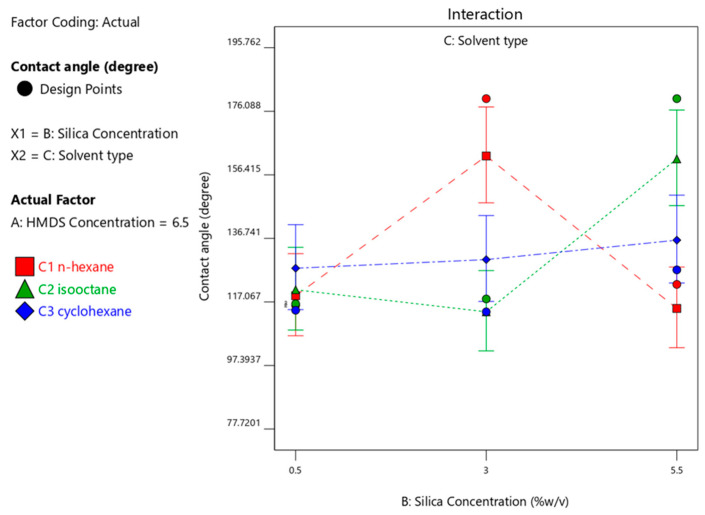
Contact angle response upon interaction between silica concentration and solvent type with 6.5%-*v*/*v* HMDS.

**Figure 10 materials-14-00530-f010:**
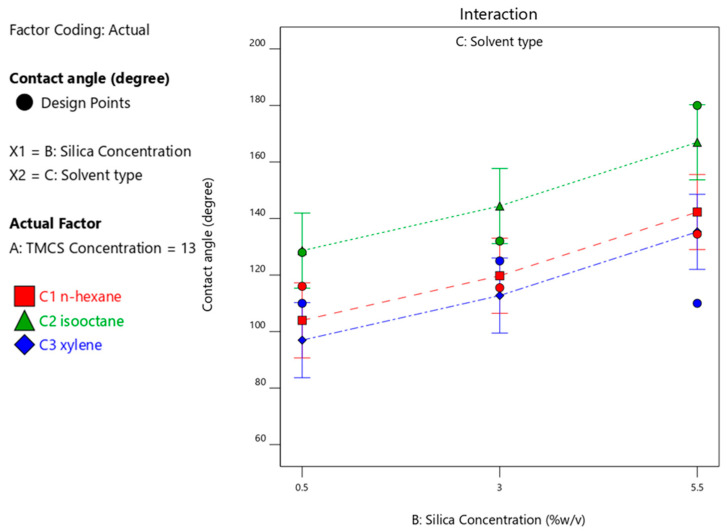
Contact angle response upon interaction between silica concentration and solvent type with 13%-*v*/*v* TMCS.

**Figure 11 materials-14-00530-f011:**
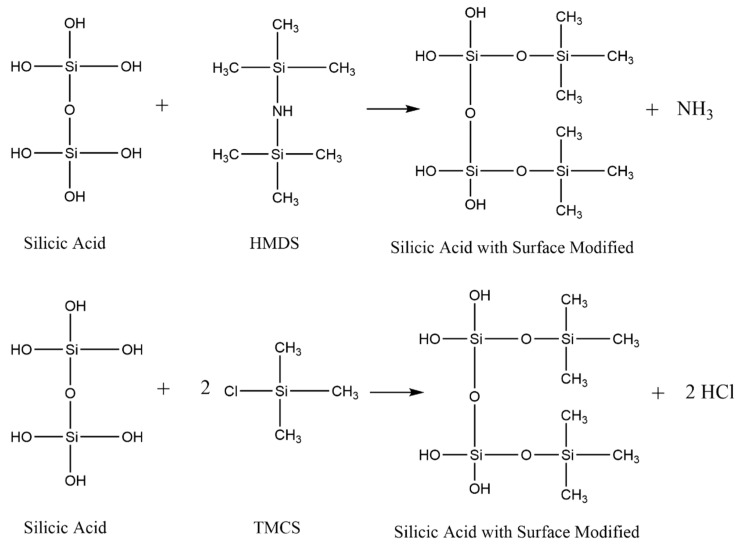
Formation reaction using HMDS and TMCS [28].

**Figure 12 materials-14-00530-f012:**
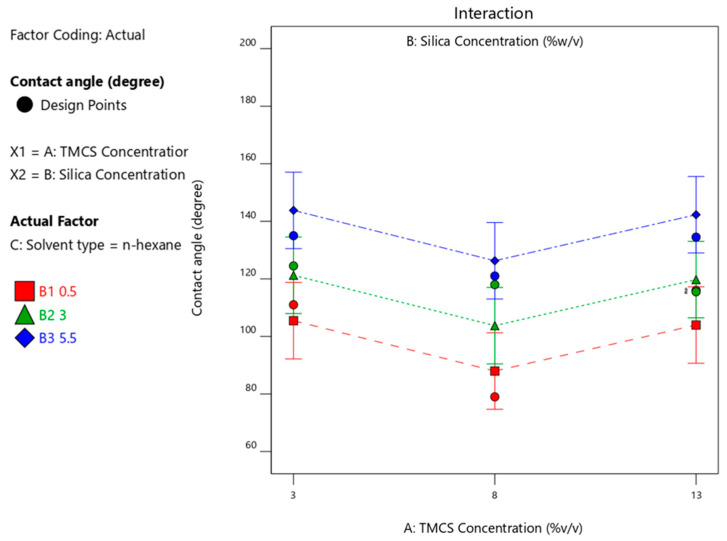
Contact angle upon interaction between TMCS and silica concentration with n-hexane solvent.

**Figure 13 materials-14-00530-f013:**
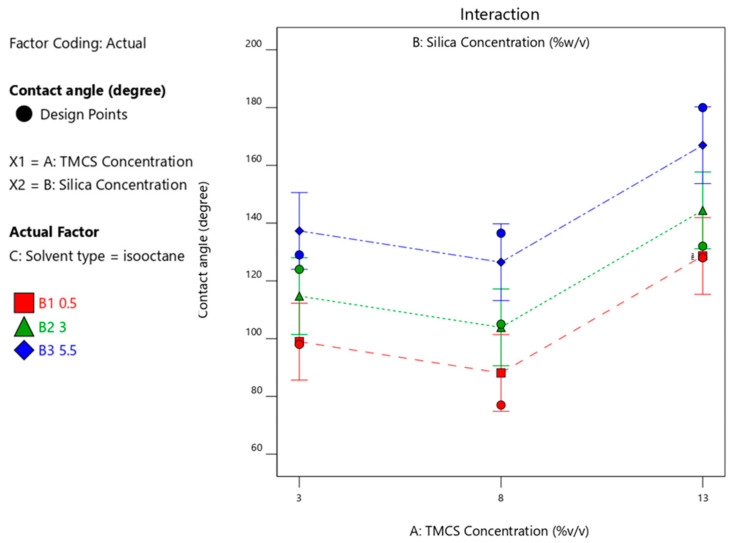
Contact angle upon interaction between TMCS and silica concentration with isooctane solvent.

**Figure 14 materials-14-00530-f014:**
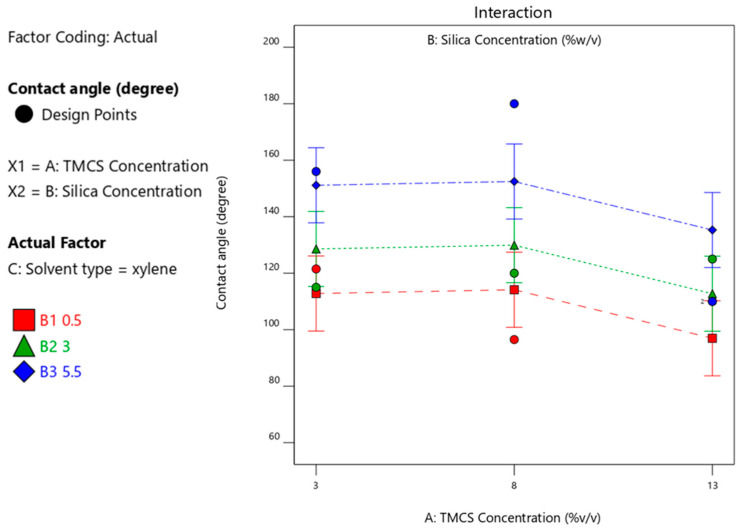
Contact angle upon interaction between TMCS and silica concentration with xylene solvent.

**Figure 15 materials-14-00530-f015:**
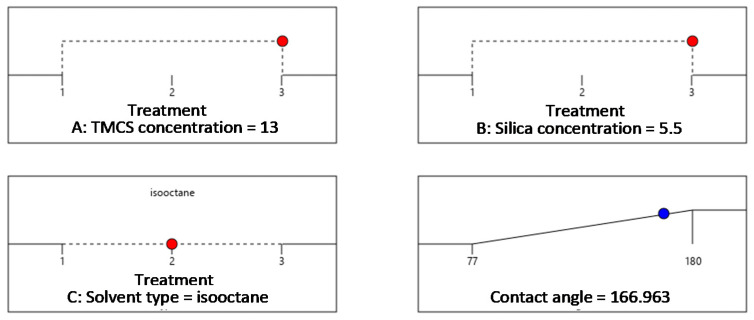
Optimization of the operating conditions of superhydrophobic coating using TMCS.

**Figure 16 materials-14-00530-f016:**
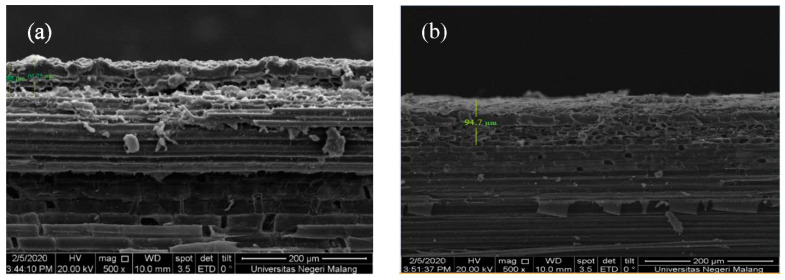
Thickness of coated bamboo with modified silica–HMDS coating with contact angles of 112.5° (**a**) and 180° (**b**).

**Figure 17 materials-14-00530-f017:**
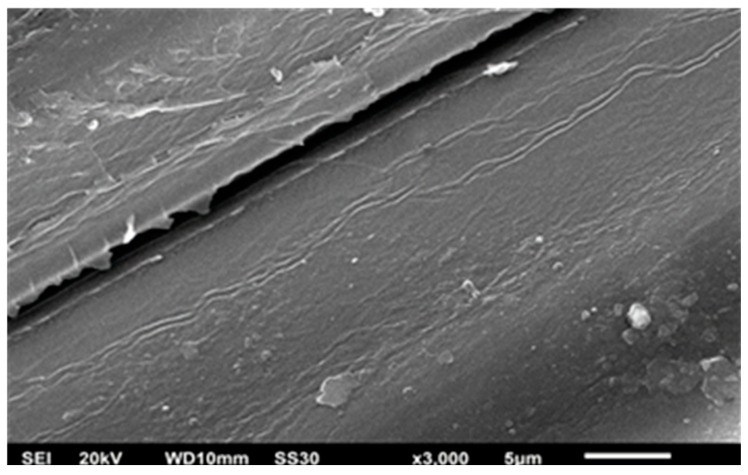
Surface morphology of uncoated bamboo.

**Figure 18 materials-14-00530-f018:**
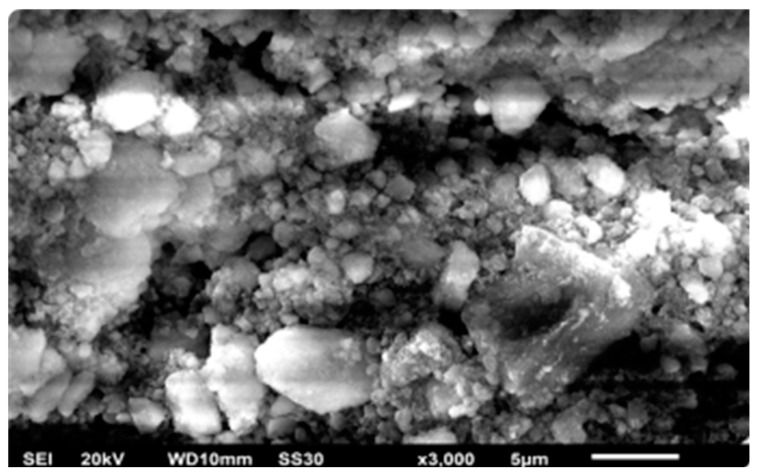
Surface morphology after superhydrophobic coating (modified silica with TMCS).

**Figure 19 materials-14-00530-f019:**
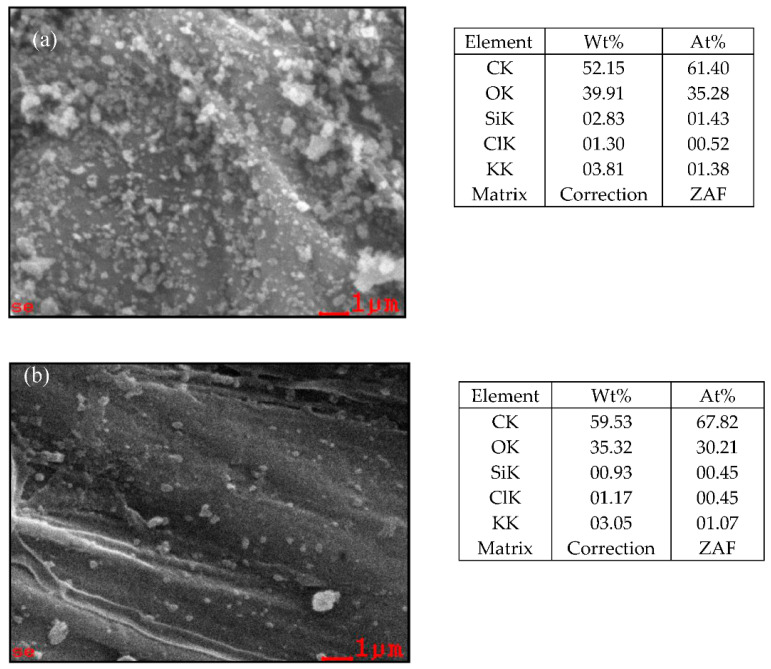
SEM-EDX mapping of coated bamboo: (**a**) n-hexane, TMCS (13%), silica (0.5%) with 129.5° contact angle; (**b**) iso-octane, TMCS (13%), silica (0.5%) with 79° contact angle.

**Figure 20 materials-14-00530-f020:**
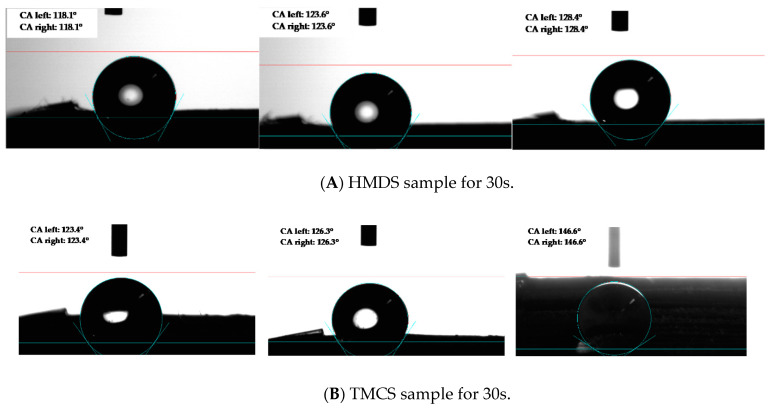
Contact angle of sample after the sand abrasion test for 30 s (**A**) HMDS and (**B**) TMCS at mounted 45°.

**Table 1 materials-14-00530-t001:** Variables experiment for full factorial design of 3 levels.

Surface-Modifying Agent	Concentration (%-*v*/*v*)	Silica Concentration (%-*w*/*v*)	Solvent Type (Categorical Variables)
HMDS	1.5	0.5	cyclohexane
	4	3	iso-octane
	6	5.5	n-hexane
TMCS	3	0.5	n-hexane
	8	3	xylene
	13	5.5	iso-octane

**Table 2 materials-14-00530-t002:** Component analysis of unleached and leached silica.

Element	Prior Treatment (%)	Post Treatment (%)
Si	86.30	97.10
Mn	0.09	0.02
Fe	3.59	0.45
Ca	3.21	0.51

**Table 3 materials-14-00530-t003:** Contact angle response with used hexamethyldisilazane (HMDS).

Run	HDMS (%-*v*/*v)*	Silica Concentration (%-*w*/*v)*	Solvent Type	Contact Angle Degree (Average of 3 Measurements)
1	4	5.5	cyclohexane	122
2	4	0.5	cyclohexane	123
3	4	3	cyclohexane	122.5
4	1.5	3	cyclohexane	128.5
5	1.5	0.5	isooctane	127.5
6	1.5	0.5	cyclohexane	124
7	1.5	0.5	n-hexane	121
8	6.5	3	cyclohexane	114
9	4	0.5	isooctane	122.5
10	4	5.5	n-hexane	123.5
11	1.5	5.5	n-hexane	142
12	1.5	5.5	isooctane	135
13	6.5	5.5	cyclohexane	127
14	1.5	5.5	cyclohexane	132.5
15	4	5.5	isooctane	125
16	6.5	3	n-hexane	180
17	4	0.5	n-hexane	111.5
18	6.5	0.5	isooctane	116.5
19	6.5	5.5	isooctane	180
20	6.5	0.5	cyclohexane	114.5
21	6.5	3	isooctane	118
22	4	3	n-hexane	121.5
23	6.5	5.5	n-hexane	122.5
24	6.5	0.5	n-hexane	116
25	1.5	3	isooctane	180
26	1.5	3	n-hexane	119
27	4	3	isooctane	112.5

**Table 4 materials-14-00530-t004:** Contact angle response with trimethylchlorosilane (TMCS).

Run	TMCS (%*v*/*v)*	Silica (%-*w*/*v)*	Solvent Type	Contact Angle Degree (Average of 3 Measurements)
1	13	0.5	n-hexane	116
2	8	0.5	xylene	96.5
3	13	5.5	isooctane	180
4	3	0.5	isooctane	98
5	3	0.5	xylene	121.5
6	8	5.5	n-hexane	121
7	8	3	n-hexane	118
8	8	5.5	xylene	180
9	3	5.5	n-hexane	135
10	8	3	xylene	120
11	13	3	n-hexane	115.5
12	3	3	isooctane	124
13	8	3	isooctane	105
14	3	3	xylene	115
15	8	0.5	n-hexane	79
16	8	0.5	isooctane	77
17	13	5.5	n-hexane	134.5
18	13	3	xylene	125
19	13	0.5	isooctane	128
20	13	5.5	xylene	110
21	13	3	isooctane	132
22	3	3	n-hexane	124.5
23	3	5.5	xylene	156
24	8	5.5	isooctane	136.5
25	13	0.5	xylene	110
26	3	0.5	n-hexane	111
27	3	5.5	isooctane	129

**Table 5 materials-14-00530-t005:** Polarity of solvents.

Solvents	Polarity Index
n-Hexane	0.1
Isooctane	0.1
Cyclohexane	0.2
Xylene	2.5

**Table 6 materials-14-00530-t006:** Durability test result on bamboo samples with silica–TMCS coating.

Time (week)	Mass (g)		Moisture Content (%)	
	Coated 180°	Uncoated	Coated 180°	Uncoated
0	9.18	8.89	0	0
10	10.05	22.65	26	28

**Table 7 materials-14-00530-t007:** Durability test result on bamboo samples with silica–HMDS coating.

Time (week)	Mass (g)			Moisture Content (%)		
	Coated (180°)	Uncoated	Coated (112.5°)	Coated (180°)	Uncoated	Coated (112.5°)
0	9.09	9.08	3.48	0	0	0
10	9.33	17.01	3.93	19	40	21

**Table 8 materials-14-00530-t008:** Contact angle (CA) for the sand abrasion at an angle of 45°.

Samples	30 s (Average of 3 Measurements)	120 s (Average of 3 Measurements)
HMDS	123.4°	120.7°
TMCS	132°	124.2°

## Data Availability

Data sharing not applicable.

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
