# Peer review of "Superhydrophobic Coating Derived from Geothermal Silica to Enhance Material Durability of Bamboo Using Hexadimethylsilazane (HMDS) and Trimethylchlorosilane (TMCS)"

_materials, 2021, doi:10.3390/ma14030530_

Round 1

Reviewer 1 Report

This research reported superhydrophobic coating derived from geothermal silica. The results are interesting. I suggested this research to be published after some modifications.

  1. The surface roughness of the sample had better be provided by the AFM test. Only SEM observation is not enough.
  2. The mechanical robustness of the superhydrophobic coating is very crucial for the practical application. If the mechanical robustness is strong in this research, the sandpaper-abrasion test is recommended. If the mechanical robustness is relatively low, the sand impact test is recommended.
  3. The multi-fluorination strategy can improve the robustness of self-stratifying coating. This strategy had better be mentioned in the introduction section (Nature Materials, 2018, 17, 355-360; Journal of Materials Chemistry A, 2020, 8, 3509-3516).
  4. The XPS test had better be performed to investigate the chemical composition.
  5. The quality of Figure 4,5,6,7,9,10,12,13,14,15 are very poor. I can not see them clearly.

Reviewer 2 Report

The authors demonstrate the fabrication of a superhydrophobic nano-coating which is applied on Bamboo. The manuscript is well written and I believe will be interesting for the readers of 'Materials'. I have only one comment, the authors should increase the analysis of figs 4, 5, 6, 7, 9, 10, 11, 12, 13, 14, 15.

Reviewer 3 Report

The present paper reports the preparation of two superhydrophobic coatings, of hexamethyldisilazane (HMDS) and trimethylchlorosilane (TMCS) solution, by spray coating technique on bamboo.
The paper presents an interesting and uncommon problem concerning a material not very widespread except in some areas of the world and the use of geothermal silica.

Nevertheless, this point, the paper in my opinion present some problem. First, the authors assess that TEOS is more harmful with respect to TMCS and HDMS but they don't support this affirmation with proper references and furthermore the safety data sheet of TMCS and HDMS report as Signal word: Danger for TMCS and HDMS and Warning for TEOS.
After this point, I observed that the preparation method of the coating by spray is not described in detail (instrument and operational conditions).
The importance and the advantage in the use of silica coming from waste is not clear and moreover, it is not mentioned in the abstract and only minimally in the introduction.

Is not my filed of experience but I don't understand very well the Design of the Experiment, in particular, the choice of the HMDS/TMCS and silica concentration. Reading the article for me is difficult to understand why to much space has been dedicated to the DeO. Furthermore, the understanding of the graphs is very difficult both for the low quality of the images and for the similarity between them. In particular, I have not been able to understand figures 7 and 14, the caption is not sufficient and the support that these images should give is not well explained in the text.
In my opinion, more space should be dedicated to field experimentation, or tests that simulate it, given that the title of the paper reports an extremely practical problem. For example the authors in fig 19 report two SEM images of two different coatings: no data of thickness and roughness were reported in the text although the authors speak of these two concepts as to justify their results. Could be more interesting and fundamental a proper and accurate surface characterization, mandatory in superhydrophobic coating field.

The paper have some flaws, if the authors will be able to improve the experiment the paper could be published.

The grammar needs to be checked. The references are scare and the authors can look at this. Furthermore I suggest to improve the contents of the introduction about superhydrophobicity, theory and state of art.

Round 2

Reviewer 1 Report

The authors have carefully modified the manuscript.

Author Response

Dear Reviewer,

Thank you for your comment.

We have already recheck all sentences. Please find revised manuscript attached in resubmitted manuscript.

Thank you for your next consideration.

Best regards,

S. Silviana

Reviewer 3 Report

The paper in the present and revised form is better than the first version, it is more clear.

Nevertheless are present some flaws and in my opinion, most pictures are not clear. In particular:

  • Figure 4, 5, 6, 9, 10, 12, 13, 14: unreadable legend
  • Figure 7: the last 4 graphs have a low quality with respect to the previous 4,
  • Figure 15: low quality
  • Figure 20: low quality, it's impossible to read the marker,
  • Figure 21: low quality, is not possible to understand the contact angle.

The paper before publication needs to be carefully read and correct because some times, in particular the new sentences, are not clear such as at line 508: After the sand abrasion test, the superhydrophobic samples altered the hydrophobic samples affirmed by decrease of the contact angle.

After a careful, meticulous check and correction of the figures to make them publishable in my opinion the paper can be published.

Author Response

Dear Reviewer,

Thank you for your comments.

Please find attached in this section of our response to your comments.

Thank you in advance for your next consideration.

Best regards,

S. Silviana
